# Oxidative Stress, Inflammation and Altered Glucose Metabolism Contribute to the Retinal Phenotype in the Choroideremia Zebrafish

**DOI:** 10.3390/antiox13121587

**Published:** 2024-12-23

**Authors:** Cécile Méjécase, Neelima Nair, Hajrah Sarkar, Pablo Soro-Barrio, Maria Toms, Sophia Halliday, Katy Linkens, Natalia Jaroszynska, Constance Maurer, Nicholas Owen, Mariya Moosajee

**Affiliations:** 1Development, Ageing and Disease, UCL Institute of Ophthalmology, London EC1V 9EL, UK; smgxcme@ucl.ac.uk (C.M.); neelima.nair@ucl.ac.uk (N.N.); h.sarkar@ucl.ac.uk (H.S.); maria.toms@crick.ac.uk (M.T.); katy.linkens.13@ucl.ac.uk (K.L.); n.jaroszynska@ucl.ac.uk (N.J.); constance.maurer.15@ucl.ac.uk (C.M.); n.owen@ucl.ac.uk (N.O.); 2Ocular Genomics and Therapeutics, The Francis Crick Institute, London NW1 1AT, UK; 3Bioinformatics and Biostatistics Science Technology Platform, The Francis Crick Institute, London NW1 1AT, UK; 4Great Ormond Street Hospital for Children NHS Foundation Trust, London WC1N 9JH, UK; 5Department of Genetics, Moorfields Eye Hospital NHS Foundation Trust, London EC1V 2PD, UK

**Keywords:** CHM, choroideremia, zebrafish, oxidative stress, glucose metabolism, inherited retinal diseases

## Abstract

Reactive oxygen species (ROS) within the retina play a key role in maintaining function and cell survival. However, excessive ROS can lead to oxidative stress, inducing dysregulation of metabolic and inflammatory pathways. The *chm^ru848^* zebrafish models choroideremia (CHM), an X-linked chorioretinal dystrophy, which predominantly affects the photoreceptors, retinal pigment epithelium (RPE), and choroid. In this study, we examined the transcriptomic signature of the *chm^ru848^* zebrafish retina to reveal the upregulation of cytokine pathways and glia migration, upregulation of oxidative, ER stress and apoptosis markers, and the dysregulation of glucose metabolism with the downregulation of glycolysis and the upregulation of the oxidative phase of the pentose phosphate pathway. Glucose uptake was impaired in the *chm^ru848^* retina using the 2-NBDG glucose uptake assay. Following the overexpression of human *PFKM*, partial rescue was seen with the preservation of photoreceptors and RPE and increased glucose uptake, but without modifying glycolysis and oxidative stress markers. Therapies targeting glucose metabolism in CHM may represent a potential remedial approach.

## 1. Introduction

The retina is a highly metabolically active tissue with the photoreceptors being one of the most energy consuming cell types of the body; a single mouse rod photoreceptor consumes approximately 10^8^ molecules of ATP per second in the dark [1]. The main energy source for photoreceptors is glucose, which is transported from the choriocapillaris to the retinal pigment epithelium (RPE) and then to the photoreceptors via glucose transporter 1 (GLUT1) [2]. In photoreceptors, glucose metabolism begins with glycolysis, a series of enzymatic reactions that convert glucose to pyruvate. In most cells, pyruvate proceeds through oxidative phosphorylation; however, photoreceptors favour aerobic glycolysis with over 80% of the glucose consumed by the retina converted to lactate in the presence of oxygen [3,4,5]. A strict regulation of retinal glucose concentration, transport and normal glycolysis are required to maintain photoreceptor health and function; an increase or decrease in glucose level can result in aberrant ERG responses [6]. *Glut1* deletion in RPE reduces retinal glucose concentration and photoreceptor survival [7]. Disruption of glycolysis enzymes, like HK2 and PKM, affect photoreceptor function and survival [8]. Of note, in physiological conditions, isomerisation of chromophore 11-*cis*-retinal by light produces byproducts of the retinoid cycle, such as lipofuscin, known to generate reactive oxygen species (ROS), and thus induces oxidative stress through accumulation [9]. Glucose anabolism via the pentose phosphate pathway reduces ROS and protects photoreceptors and RPE from oxidative stress-induced apoptosis [9,10]. However, the correct processing of ROS is important to avoid compromising cell function and survival [11]: uncontrolled oxidative stress can lead to glucose metabolism dysregulation, ER stress, inflammation, and cell death. Oxidative stress-induced glucose metabolism creates a positive feedback loop, amplifying oxidative stress and cell degeneration.

Choroideremia (CHM) is an X-linked chorioretinal dystrophy caused by mutations in the *CHM* gene, affecting the photoreceptors, retinal pigment epithelium (RPE), and choroid. CHM affects male patients, and it is characterised by night blindness in early childhood, with progressive constriction of the visual field and eventual loss of central vision leading to complete blindness by the fifth decade of life [12]. Female carriers are mostly asymptomatic; some can experience night blindness in middle age, and fewer being more severely affected similar to male patients. A fundus examination in male CHM patients reveals chorioretinal atrophy, which begins in the periphery with scalloped edges and later coalesces so that the sclera can be seen. The macula is spared with a small retinal island until the fifth to sixth decade of life. The well-characterised *chm^ru848^* zebrafish model of CHM displays a rapid and progressive retinal degenerative phenotype, with early signs of defective vasculogenesis of the choriocapillaris, photoreceptor cell loss and hypertrophic RPE at 4 days post fertilisation (dpf), followed by loss of retinal lamination and widespread cell death by 5 dpf [13,14]. The ubiquitously expressed zebrafish *chm* gene encodes rab escort protein 1 (rep1), which is involved in lipid prenylation of rab proteins and intracellular trafficking; zebrafish do not have a REP2 orthologue as in humans, hence no compensation for loss of *chm* is seen, resulting in systemic degeneration and embryonic lethality by mean 5 dpf. Previous studies have shown oxidative and ER stress in the *chm^ru848^* zebrafish retina [14]. Hence, this model was chosen to further investigate ROS processing through transcriptomic analysis, which implicated inflammatory pathways, cellular stress, apoptosis, and glucose metabolism. Phosphofructokinase (PFK) is a key enzyme of the glycolysis pathway and catalyses the phosphorylation of fructose-6-phosphate (F6P) to fructose-1,6-bisphosphate (F-1,6-BP). We performed a proof-of-concept study, investigating whether enhancing glucose uptake in the photoreceptors, via overexpression of the human *PFKM* isoform, can maintain cell activity and promote photoreceptor survival.

## 2. Materials and Methods

### 2.1. Zebrafish Husbandry

Wild-type AB (wt) and choroideremia (*chm^ru848^*) zebrafish were bred and maintained according to local UCL and UK Home Office regulations for the care and use of laboratory animals under the Animals Scientific Procedures Act at the UCL Bloomsbury campus zebrafish facility. Zebrafish were raised at 28.5 °C on a 14 h light/10 h dark cycle. The UCL Animal Welfare and Ethical Review Body approved all procedures for experimental protocols, in addition to the UK Home Office (License no. PPL PC916FDE7 and PP0390240). All approved standard protocols followed the guidelines of the ARVO Statement for the Use of Animals in Ophthalmic and Vision Research Ethics.

### 2.2. Transcriptomic Signature Analysis

Bulk RNA Seq was performed using high-quality RNA isolated from *chm^ru848^* and wt control retinas, RPE included (*n* = 7 zebrafish per group) [15]. cDNA libraries were subsequently constructed from total RNA (RIN ≥ 8) using the Clontech SMART-Seq v4 Ultra Low Input RNA Kit and sequenced at Novogene (UK). Raw reads were quality and adapter trimmed using trimgalore (version 0.6.7) [16] before alignment. Reads were mapped and subsequent gene-level counted using RSEM 1.3.3 [17] and STAR 2.7.10a [18] against the zebrafish genome GRCz11, both from Ensembl, using the nf-core/rna-seq pipeline (version 3.10.1) [19,20]. Normalisation of raw count data and differential expression analysis was performed with the DESeq2 package (version 1.38.3) [21] within the R programming environment (version 4.2.2) [22]. Differentially expressed genes between chm vs. wt were determined with the contrast function of a pairwise comparison, with the adjusted *p* value threshold at 0.05. Gene lists were used to look for pathways and molecular functions with over-representation analysis using Gene Ontology (GO) [23]. Heatmaps were created with ComplexHeatmap package (version 2.14.0) [24].

### 2.3. Data Availability

The RNA-seq data generated by this study have been deposited in the NCBI Gene Expression Omnibus under the access code GSE254948, including unprocessed FASTQ files and associated gene count matrices.

### 2.4. PFKM Overexpression in Zebrafish Models

Human *PFKM* with 5′ BamHI and 3′ XhoI restriction sites was purchased as a genestring from Thermo Fisher (Waltham, MA, USA), restriction digested and cloned into pCS2+ vector using T4 DNA ligase (New England Biolabs, Ipswich, MA, USA). Plasmid was transformed into OneShot TOP10 chemically competent *E. coli* cells (Thermo Fisher) and purified using QIAprep Spin Miniprep kit (Qiagen, Hilden, Germany), according to the manufacturer’s instructions. Insertion of the *PFKM* sequence was confirmed by Sanger sequencing. Plasmid was linearized with NotI restriction enzyme, cleaned up, and transcribed using the mMessage mMachine SP6 kit (Thermo Fisher), followed by purification using the RNeasy Mini Kit (QIAGEN), according to the manufacturer’s instructions. Doses between 100 and 400 ng/μL were tested by injection directly into 1–2 cell stage wt embryos. The injection solution was prepared by diluting with nuclease-free H_2_O and adding 1:10 phenol-red; 2 nL was injected into each embryo. Survival curve was produced after at least 3 repeats per dose. A dose of 200 ng/μL *PFKM* was found to be optimum without causing toxicity or death.

### 2.5. Zebrafish Characterisation

Embryos at 5 dpf were fixed in 4% paraformaldehyde (PFA) overnight at 4 °C and embedded using the JB-4 embedding kit (Polysciences Inc., Warrington, PA, USA), according to the manufacturer’s instructions. Sections were cut at a thickness of 7 μm, stained with 1% toluidine blue and imaged on an a Axioplan 2 microscope (ZEISS Microscopy, Jena, Germany). A TUNEL assay was performed on cryosections using the ApopTag Fluorescein In Situ Apoptosis Detection Kit (Millipore, Burlington, MA, USA) according to the manufacturer’s instructions. Slides were counterstained and mounted with Prolong Diamond Antifade Mountant  +  DAPI and imaged on a Zeiss LSM710 upright confocal microscope. TUNEL positive cells were counted using ImageJ software (version 2.14/1.54f) [25].

### 2.6. 2-NBDG Glucose Uptake Assay

Embryos at 5 dpf were placed individually in a 96 well plate and incubated with 600 μM 2-NBDG (Thermo Fisher Scientific) in E3 medium for 3 h. Fish were fixed in 4% PFA overnight at 4 °C and processed for cryosectioning as above. Slides were permeabilised in 0.01% Triton-X for 20 min and then fixed in 4% PFA for 10 min before mounting with Prolong Diamond Antifade Mountant  +  DAPI and imaged on a Zeiss LSM710 upright confocal microscope. The fluorescence intensity was measured using ImageJ software. The area to be measured was selected and the integrated density was measured. The corrected total cell fluorescence (CTCF) was calculated with the difference between measured integrated density, and the area of the selected region was multiplied by the mean of the fluorescence background.

### 2.7. RT-qPCR

The total RNA was extracted from pools of 20 zebrafish eyes using the RNeasy FFPE kit (Qiagen). cDNA was synthesised from 1 μg of RNA using qScript cDNA supermix (Quantabio, Beverly, MA, USA) according to the manufacturer’s instructions. mRNA levels were analysed using SYBR Select Master Mix (Applied Biosystems, Waltham, MA, USA) on a StepOne Real-Time PCR system (Applied Biosystems) under standard cycling conditions. The results were normalised to housekeeping gene B-actin. All samples were assayed in triplicates. The primer sequences for zebrafish apoptosis, glycolysis, pentose phosphate, and oxidative stress genes are shown in Table 1.

### 2.8. Statistical Analysis

All other statistical analyses were performed using GraphPad Prism 9 and data are expressed as mean ± SEM. *p* value of ≤ 0.05 was considered significant.

## 3. Results

### 3.1. RNAseq Analysis Reveals Disrupted Glucose Metabolism in chm^ru848^ Fish, Associated with Increased Inflammatory Responses

The *chm^ru848^* zebrafish retina is known to share comparable phenotypic features to CHM patients [26]; however, much of the disease aetiology remains unknown. To unravel disease-mechanisms associated with CHM, RNAseq analysis was performed on dissected retinas from wt and *chm^ru848^* zebrafish at 5 dpf, which revealed a clear differential transcriptomic signature: 4445 genes were downregulated while 4425 genes were upregulated in the *chm^ru848^* model. Interestingly, gene-ontology enrichment revealed immune responses (including GO:0002253, GO:0060326, GO:0070098, GO:0071347, GO:0019221, GO:0006954), transcription and translation (including GO:0006412, GO:0008033, GO:0050658, GO:0006397, GO:0034470), visual perception (including GO:0007601, GO:0007602), glucose metabolism (including GO:0006112, GO:0044042, GO:0005977, GO:0005978, GO:0043467), oxidative stress (including GO:0042554), ER stress (including GO:0034620, GO:0034976) and apoptosis (including GO:0010942, GO:0010941) are highly perturbed in *chm^ru848^* fish (Appendix A).

Interestingly, amongst differential genes associated with an immune response, cytokines genes and associated pathways were upregulated, including *myd88* (LFC 1.441, padj < 1.434 × 10^−3^), *il34* (LFC 2.000, padj < 1.367 × 10^−5^), *il1b* (LFC 3.057, padj < 7.987 × 10^−3^), *tyk2* (LFC 3.679, padj < 5.643 × 10^−17^), *irak3* (LFC 4.055, padj < 4.783 × 10^−10^), *cxcl8a* (LFC 4.867, padj < 1.589 × 10^−5^), *il11a* (LFC 6.384, padj < 6.752 × 10^−8^), *il12a* (LFC 8.872, padj < 1.497 × 10^−73^) and *il11b* (LFC 9.466, padj < 3.099 × 10^−28^) (Figure 1 and Appendix A).

Previous work showed dysregulation of ER and oxidative stress markers in *chm^ru848^* retina [14]. In the present study, transcriptomic analysis showed increased oxidative stress with the downregulation of antioxidant enzymes (*sod3a* and *cat*) and the upregulation of proteins involved in ROS production (*cyba*, *nox1*, *noxo1b*, *ncf4* and *noxo1a*) (Figure 2a and Appendix A). Unfolded protein response to ER stress (*atf6*, *atf4a*, *dnajc10*, *hspa5*, *hsp70.3*, *xbp1*) and proteasome complex (*psme3*, *psme2*, *psme1*) genes were upregulated in *chm^ru848^* zebrafish (Figure 2b and Appendix A). Apoptosis makers were upregulated in mutant *chm^ru848^* zebrafish retina: *casp3a* (LFC 0.805, padj < 1.623 × 10^−6^), *casp8* (LFC 1.077, padj < 3.566 × 10^−2^), *bcl10* (LFC 1.620, padj < 2.312 × 10^−3^), *tp53* (LFC 1.695, padj < 1.979 × 10^−14^), *casp3b* (LFC 1.701, padj < 4.094 × 10^−2^), *baxb* (LFC 1.806, padj < 3.378 × 10^−4^), *baxa* (LFC 1.912, padj < 2.333 × 10^−14^) and *ddit3* (LFC 2.580, padj < 1.685 × 10^−21^) (Figure 2c and Appendix A).

GLUT transporters were significantly dysregulated: *slc2a1a* (LFC −3.873, padj < 2.222 × 10^−6^) and *slc2a3a* (LFC −1.439, padj < 4.986 × 10^−6^) were downregulated; and *slc2a1b* (LFC 1.535, padj < 1.587 × 10^−2^) and *slc2a3b* (LFC 2.900, padj < 1.069 × 10^−8^) were upregulated. Glucose hexokinases were significantly upregulated (*hk1*, *hk2* and *gck*); while genes involved in glycolysis downstream reactions were downregulated in *chm^ru848^* retina (Figure 3a,b with upregulated genes in red and downregulated in blue, adapted from [27] and Appendix A). Interestingly, the pentose phosphate pathway oxidative branch were upregulated: *pgd* (LFC 2.567, padj < 2.348 × 10^−12^), *h6pd* (LFC 2.137, padj < 8.300 × 10^−12^), *rpe* (LFC 1.235, padj < 7.498 × 10^−5^), *prps1a* (LFC 1.15, padj < 8.461 × 10^−7^), *prps1b* (LFC 0.819, padj < 2.023 × 10^−2^); and the non-oxidative branch were downregulated: *tkta* (LFC −2.039, padj < 8.332 × 10^−8^) (Figure 3c and Appendix A).

### 3.2. Human PFKM Overexpression Improves Retinal Phenotype

The retinal phenotype of the *chm^ru848^* zebrafish at 5 dpf showed a loss of lamination, widespread pyknotic nuclei from cell death, RPE hypertrophy and atrophy, and a small cataractous lens (Figure 4a). In order to investigate whether enhancing glucose metabolism improves photoreceptor survival, we delivered human *PFKM*, encoding a key glycolytic pathway enzyme into *chm^ru848^* zebrafish at the 1–2 cell stage. Human *PFKM* expression was seen at 5 dpf in injected zebrafish (Figure 4a). At 5 dpf, injected and uninjected wt controls showed normal retinal lamination and no sign of toxicity or cell death (Figure 4b,c). The phenotype in *PFKM*-treated *chm^ru848^* mutant embryos showed partial rescue at 5 dpf with a largely intact photoreceptor layer and thicker RPE compared to the uninjected mutants. TUNEL assay at 5 dpf in *chm^ru848^* mutant embryos showed high levels of apoptosis in the retina (Figure 4c). *PFKM* injection in *chm^ru848^* mutants did not reduce the number of apoptotic cells. RT-qPCR analysis confirmed the RNAseq results, showing the upregulation of the anti-apoptotic marker *mcl-1a* and the upregulation of the pro-apoptotic marker *ddit3* in *chm^ru848^* zebrafish (Figure 4d,e and Appendix A). While *cflara* expression remained unchanged after treatment, but pro-apoptotic markers *casp8* and *ddit3* were upregulated with the partial restoration of *mcl-1a* expression.

### 3.3. PFKM Increases Glucose Uptake in chm^ru848^ Zebrafish

Glucose uptake into live cells can be monitored using 2-NBDG (2-(N-(7-Nitrobenz-2-oxa-1,3-diazol-4-yl)Amino)-2-Deoxyglucose), a fluorescent glucose analogue [29]. Uninjected and injected wt zebrafish showed 2-NBDG staining in the photoreceptors (Figure 5). Untreated *chm^ru848^* revealed an impairment of glucose uptake into the photoreceptors compared to wildtype embryos. An injection of *PFKM* mRNA in *chm^ru848^* mutant rescued the glucose uptake in the photoreceptor cells.

### 3.4. PFKM Overexpression Does Not Improve Activity of the Glycolysis Pathway or Oxidative Stress

To understand the effect of *PFKM* overexpression on the glycolysis pathway, mRNA expression of glycolysis markers were studied at 5 dpf for significantly dysregulated genes in *chm^ru848^* transcriptomic analysis, including *hk2*, *pfkma*, *gapdh*, *and slc2a1b* (Figure 6a). RT-qPCR analysis confirmed that *hk2* is significantly upregulated and *pfkma* significantly downregulated in uninjected *chm^ru848^* zebrafish compared to wt, with no significant difference in *gapdh and slc2a1b*. *PFKM* overexpression in 1–2 cell stage embryos led to a significant upregulation of *gapdh* in wt at 5 dpf compared to uninjected wt, but no significant change in injected *chm^ru848^* zebrafish compared to uninjected mutants. There was no change in the expression of *hk2*, *pfkma*, and *slc2a1b* after *PFKM* overexpression in *chm^ru848^* zebrafish. Glucose anabolism via the pentose phosphate pathway oxidative branch was studied at 5 dpf: RT-qPCR analysis confirmed that *pgd*, *prps1a*, and *prps1b* are significantly upregulated in uninjected *chm^ru848^* compared to wt (Figure 6b). *PFKM* overexpression in *chm^ru848^* resulted in trends of reduced *prps1a* and *prps1b* and a significant reduction in *pgd* expression at 5 dpf. As chronic activation of the oxidative branch of the pentose phosphate pathway increased oxidative stress [11], three antioxidant enzymes, *cat*, *sod3a*, and *txn* were studied (Figure 6c). While *cat* and *sod3a* showed a reduced trend, *txn* had an increased trend in *chm^ru848^* zebrafish in RNAseq data (Figure 2a and Figure 6c and Appendix A). Injected and uninjected *chm^ru848^* have similar expression levels for the three antioxidant enzymes (Figure 6c).

## 4. Discussion

Our study reports the transcriptomic signatures in the *chm^ru848^* zebrafish retina, with a focus on the dysregulation of inflammatory responses, oxidative and ER stress, apoptosis and glucose metabolism. Human *PFKM* overexpression at an early stage of zebrafish eye development improves retinal lamination and restores glucose uptake. However, cell viability, glycolysis, pentose phosphate pathway, and oxidative stress remains unchanged in *chm^ru848^* zebrafish. This study is proof of concept that using human *PFKM* to target glucose metabolism can partially improve the *chm^ru848^* retinal phenotype.

The retina is highly metabolically active and consumes a large quantity of glucose to maintain normal function [3]. Glucose is transported through blood vessels in the choroid to the RPE, which then supplies the photoreceptor cells via GLUT transporters (GLUT1 and GLUT3) [30]. These are class I GLUT transporters, encoded by the *SLC2A* family of genes, which accelerate glucose entry into the cell and subsequent glucose phosphorylation by hexokinase and glucokinase [31,32]. In humans, GLUT1 and GLUT3 are each encoded by a single gene (*SLC2A1* and *SL2CA3*, respectively). However, in zebrafish, GLUT1 and GLUT3 orthologues are encoded, respectively, by *slc2a1a*, *slc2a1b* and *slc2a3a*, *slc2a3b* due to teleost-specific genome duplication [33]. Both *slc2a3a* and *slc2a3b* are expressed in the retina, with a higher expression of *slc2a3a* [34]; but the retinal localisation of each *slc2a1* isoform (*slc2a1a*, *slc2a1b*) remains unknown. Moreover, the function of each *slc2a1* and *slc2a3* isoform are unreported; further experiments are needed to characterise the function of each isoform, if they are similar, redundant, or differ. In the *chm^ru848^* retina, *slc2a1a* and *slc2a3a* are downregulated and *slc2a1b*, *slc2a3b* glucose transporters are upregulated compared to wt controls. These changes can be due to retinal dysfunction and degeneration of photoreceptors and RPE cells. Moreover, *slc2a1* genes and the oxidative branch of the pentose phosphate pathway regulate angiogenesis and vessel maturation [35,36]. We recently reported that choroidal vessel diameter and intussusceptive angiogenesis are significantly reduced in the *chm^ru848^* zebrafish [37], hence it is possible that the upregulation of the oxidative branch of the pentose phosphate pathway leads to increased elastin production by endothelial cells and consequently excessive mural cell recruitment and attachment around blood vessels [36,38], resulting in the choroidal vessel phenotype described.

The *chm^ru848^* retina failed to uptake the fluorescent glucose analogue, 2-NBDG. However, human *PFKM* overexpression showed a partial rescue of the retinal phenotype. Interestingly, D-glucose competitively inhibits 2-NBDG uptake [39,40,41], and this could explain the lack of 2-NBDG fluorescence noted in the *chm^ru848^* model, as it may have accumulated glucose from the persistent maternal yolk sac [13]. During embryogenesis, the larvae utilises the D-Glucose present in the maternal yolk sac, reducing glucose concentration levels by half in 3 days [42]. However, it is more likely that the lack of 2-NBDG fluorescence in the *chm^ru848^* retina reflected a degenerated photoreceptor layer with no capacity to uptake 2-NBDG. The *PFKM* overexpression facilitated glucose uptake in the *chm^ru848^* mutant, but the transport mechanism remains unknown. The *PFKM* overexpression showed a trend towards reducing the expression of pentose phosphate genes (oxidative branch), and it facilitates glycolysis in place of glucose anabolism. A *PFKM* injection was performed at the 1–2 cell stage, facilitating the survival of retinal progenitor cells during the first days of embryogenesis. Through inducing glycolysis, photoreceptors were able to develop and mature, as observed in histology images with better retinal lamination in the *chm^ru848^* injected zebrafish eyes. Moreover, the more preserved retinal lamination in the injected *chm^ru848^* zebrafish suggests that administration of *PFKM* during embryogenesis may support early choroidal development, as *PFK* genes have been previously reported to be involved in angiogenesis [43]. This may, in turn, slow the ensuing retinal degeneration by supporting the RPE and preventing photoreceptor cell death. Choroidal vessels provide oxygen and nutrients to the retina; however, in the *chm^ru848^* zebrafish, choroidal vessels diameters are significantly reduced [37]. This may lead to reduced oxygen availability in RPE, inducing retinal hypoxia and consequently perpetuating ROS production.

REP1 interacts with Rab GTPase proteins and recruits geranylgeranyl transferase type-II (GGTase-II) for their lipid prenylation [12,44], a post-translational modification of the hydrophobic carbon chains, essential for activation and correct intracellular localisation. REP1 targets includes Rab27A, Rab27B, Rab38, and Rab42 in RPE and Rab11b, Rab18, and Rab1B in photoreceptor outer segments [45,46,47]. Prenylation of these Rab proteins has been reported to be important for melanin production in RPE [37,46], phagocytosis, exocytosis, intracellular transport, and the immune response [47,48,49]. Pathogenic variants involving the *CHM* gene result in an accumulation of a subset of REP1-dependent unprenylated Rab proteins, which remain in a non-functional, mis- or unfolded state [50]. These inactive Rab proteins, observed in CHM patients’ fibroblasts and in human RPE models [50,51,52], accumulate in the ER, altering its homeostasis and activating an unfolded protein response (UPR) and the proteasome complex [53], resulting in ER stress and triggering cell death. The protein and organelle (endosome, lysosome, trafficking vesicles) mislocalisation also affects normal cellular function [50,52] and leads to (photo)-oxidative stress [14,37,54], which induces glucose metabolism by reducing glycolysis and enhancing the pentose phosphate pathway [11]. The oxidative phase of the pentose phosphate pathway produces NADPH, which increases oxidative stress via *NOX* genes [11], and ribose-5-phosphate, for nucleotide synthesis, also in response to ER stress. Glucose deprivation affects N-glycosylation [55], a post-translational modification that occurs in ER, leading to more protein misfolding and ER stress. The UPR and proteasome pathway activate the production of cytokines [56,57], and oxidative stress activates the inflammatory response [58]. Inflammation can cause microglia activation and monocyte recruitment in retinal tissue, as previously reported in some patients [59], which may further impact on cell survival [60]. These positive feedback loops from different disease pathways worsen the ability of the cell to resist multiple insults associated with the lack of rep1, and consequently may explain the severe phenotype observed in *chm^ru848^* zebrafish, where there is no compensation from a rep2 isoform. However, it is possible that the systemic degeneration may also cause release of ROS, inflammatory cytokines and immune cells that externally influence the rate of retinal degeneration, hence further studies in different CHM disease models are required to validate the findings of this study.

To advance our understanding of the disease mechanisms associated with CHM, we examined the transcriptomic signatures of the *chm^ru848^* zebrafish retina and revealed the dysregulation of glucose metabolism, oxidative and ER stress, and inflammation response. These pathways have also been implicated in several other IRDs. Glucose metabolism is highly affected in IRDs associated with variants involved in the glycolysis pathway, such as *HK1* [61,62] and *HKDC1* [63] associated with autosomal dominant or recessive RP, respectively; and in two retinal degeneration mouse models, *rd2* and *rd10*, characterised by decreased levels of GLUT1 protein [64]. A previous study in the *rd1* mouse model shows that RdCVF protein, Basigin-1, and GLUT1 mediate glucose uptake into cones, which promotes their survival [65]. Oxidative and ER stress have been reported in several RP models, such as *RDH12* [66], *RHO* [67], and *PDE6B* [68] mice models. The *Rd10* mouse and four RP canine models (rcd1, xlpra2, erd, xlpra1) are also characterised by upregulation of cytokines, including *Il1b* [69,70]. Moreover, pro-inflammatory cytokines like *il34*, upregulated in the *chm^ru848^* retina, activates microglia and induces their migration [71]. Oxidative stress, altered metabolic activity, immune activation, and inflammation are observed in age-related macular degeneration (AMD) [72]. AMD is characterised by RPE and choriocapillaris dysfunction and degeneration, with neovascularization in wet AMD. At a cellular level, oxidative stress in RPE cells cause premature senescence, and senescent cells produce ROS, which exacerbates the phenotype. They release cytokines and pro-inflammatory molecules, such as IL-1β. Moreover, oxidative stress mediates thioredoxin-interacting protein (TXNIP), which promote inflammation via NLRP3 inflammasome activation. In parallel, TXNIP inhibits the senescence activator AKT by direct interaction or by inhibiting glucose uptake. In the *chm^ru848^* retina, oxidative stress, cytokines, and *cdkn1a* (p21 pathway) are upregulated, both associated with premature senescence [73]. AMD and CHM share common features and may share common disease-causing pathways, involving premature senescence associated with a glucose metabolism defect, inflammation, ER, and oxidative stress.

There is currently no approved treatment for CHM, despite multiple late-phase AAV gene therapy clinical trials (NCT02341807, NCT03496012). Several options remain under preclinical development as non-viral gene therapies, which have shown promising results in CHM patients fibroblasts and the *chm^ru848^* zebrafish [74]. Translational readthrough inducing drugs, targeting nonsense variants, have been shown to partially rescue prenylation function in CHM patients fibroblasts and the *chm^ru848^* zebrafish [75]. The crosstalk between oxidative stress, altered metabolic activity, immune activation and inflammation provides more possible targets to treat CHM; for example, targeting neuroinflammation by inhibiting IL-1β or MYD88 pathways has shown some retinal improvement in a *rd10* mouse model [76]. Antioxidants (taurine and N-Acetylcysteine amide [NACA]) and anti-ER stress (tauroursodeoxycholic acid [TUDCA] and taurine) drugs did not show any retinal improvement in CHM patient fibroblasts or *chm^ru848^* zebrafish [14], but they still represent modifiable disease pathways. Interestingly, N-acetylcysteine (NAC) reduced oxidative stress and improve cone survival in *rd1* and *rd10* mouse models [77]. Oral NAC has reached a phase III clinical trial for RP patients (NCT05537220) [78]. *Txnip*, which is an α-arrestin that binds thioredoxin was shown to promote cone survival and visual acuity in the *rd1* mouse. The ability of *Txnip* to rescue cones has been attributed to a switch in fuel choice from glucose to lactate [79]. Insulin treatment in *Pde6b^−/−^* mouse models has been shown to delay cone photoreceptor degeneration [80], via the activation of the mTOR pathway, which regulates glycolysis [81]. In recent years, glucose metabolism has become an attractive target for the treatment of IRDs [82]. A common feature of many IRDs, such as retinitis pigmentosa (RP) and Leber congenital amaurosis (LCA), is photoreceptor cell death. This typically begins with loss of the rod photoreceptors, resulting in night blindness and then the constriction of the visual field, followed by secondary loss of cones, which eventually leads to complete blindness. Cone cell death, despite mutations in rod-specific genes, has been attributed to the loss of rod-derived cone viability factor (RdCVF), a molecule that is secreted by rods and interacts with Basigin-1 and GLUT1 to mediate glucose uptake into cones [65]. An early phase I/II clinical trial (NCT05748873) is underway for a retinal gene therapy using an adeno-associated virus (AAV) vector containing both RdCVF and RdCVF-long form (https://sparingvision.com/spvn06/, accessed on 24 January 2024). AAV-mediated delivery of RdCVF in *rd10* mice was shown to improve cone function, through increased cone density and ERG b-waves [83]. Hence, patients with pathogenic variants in *RHO*, *PDE6A* or *PDE6B* are currently being recruited for a phase I/II clinical trial for AAV-RdCVF-RdCVFL (NCT05748873). Further studies have shown that targeting the oxidative pentose phosphate pathway can be beneficial in cancer by reducing proliferation and antioxidant production [84,85]. Moreover, variants or disruption in pentose phosphate pathway genes (*G6PD*, *TKT*, *TALDO1*) induces oxidative stress and cell apoptosis [86,87,88,89,90]. Our results show that human *PFKM* overexpression causes a partial rescue of the *chm^ru848^* phenotype, hence it needs further validation in other CHM models such as a *Chm* mouse and *CHM* induced pluripotent stem cell (iPSC)-derived RPE/retinal organoids. CHM is a potential candidate for the AAV-RdCVF-RdCVFL as both *nxnl1* and *nxnl2* were significantly downregulated in the *chm^ru848^* retina, as observed in P23H RHO zebrafish model [67]. RdCVF expression might facilitate glucose uptake and glycolysis, reducing the effects of feedback loops due to the dysregulation of glucose metabolism.

## 5. Conclusions

Our study is the first to report the transcriptomic signature in the *chm^ru848^* zebrafish model and provides further insights into the pathophysiology of CHM. As CHM affects different types of cells, including RPE, photoreceptor, and choroidal cells further analysis performing single cell RNA sequencing on CHM models will be informative. This will help elucidate the involvement and degenerative processes at play and allow for new therapeutic targets to be considered in the future.

## Figures and Tables

**Figure 1 antioxidants-13-01587-f001:**
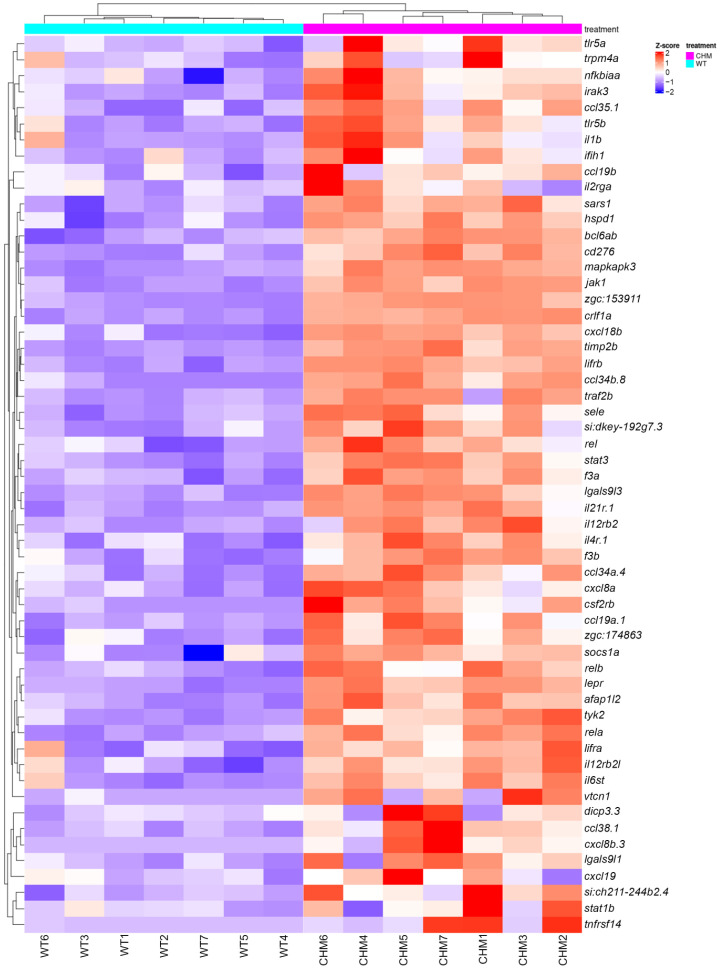
Cytokine pathway is upregulated in *chm^ru848^* zebrafish retina. *chm^ru848^* zebrafish samples are indicated in magenta and wt samples are indicated in cyan.

**Figure 2 antioxidants-13-01587-f002:**
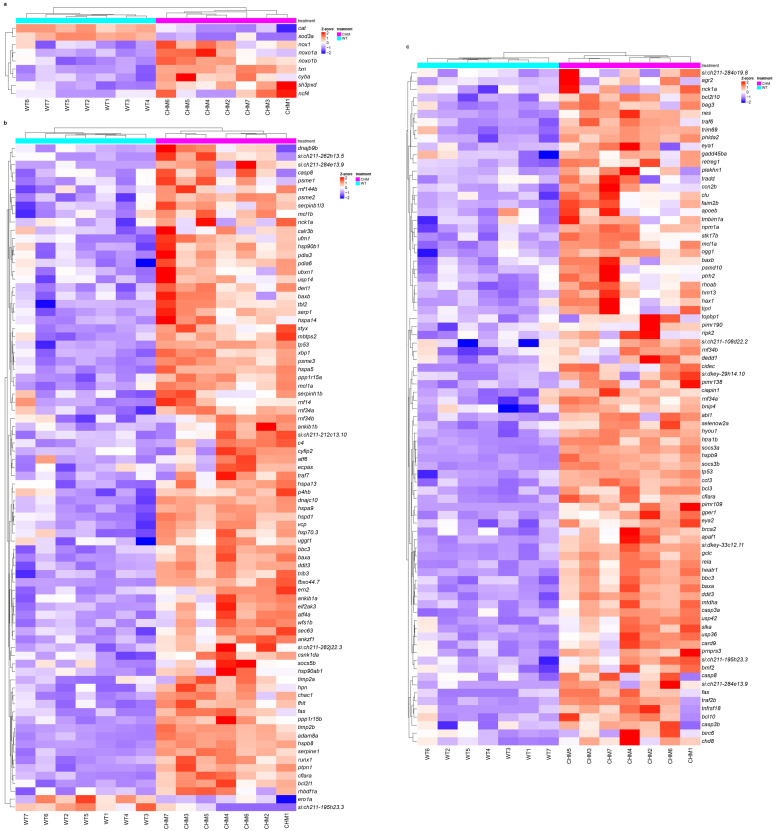
Oxidative (**a**), ER stress (**b**) and apoptosis (**c**) markers are upregulated in the *chm^ru848^* zebrafish retina. *chm^ru848^* zebrafish are indicated in magenta and wt samples are indicated in cyan.

**Figure 3 antioxidants-13-01587-f003:**
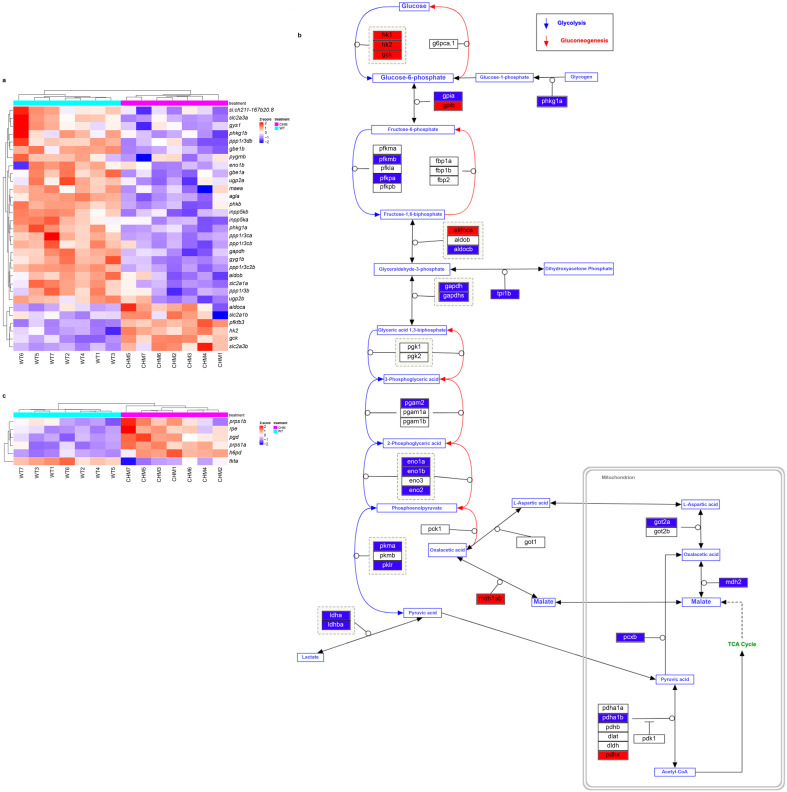
Glycolysis is downregulated in favour of the pentose phosphate pathway in *chm^ru848^* zebrafish retina. (**a**,**b**) Glycolysis pathways are affected in *chm^ru848^* retina. (**a**) *chm^ru848^* zebrafish are indicated in magenta and wt samples are indicated in cyan. (**b**) Scheme adapted from https://www.wikipathways.org/instance/WP1356 (accessed on 18 December 2024) [27,28]. Significantly downregulated genes are indicated in blue, while upregulated genes are in red. (**c**) Pentose phosphate pathway is upregulated in the *chm^ru848^* retina compared to wt.

**Figure 4 antioxidants-13-01587-f004:**
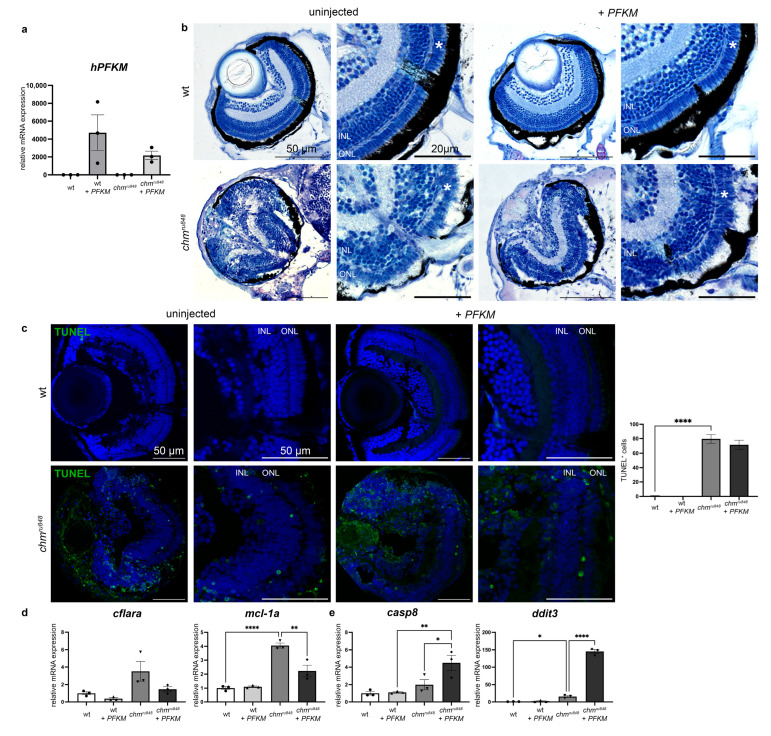
*PFKM* overexpression improves retinal phenotype but not photoreceptor cell death at 5 dpf. (**a**) Human *PFKM* expression was detected 5 days post-injection in *chm^ru848^* and wt zebrafish. (**b**) mRNA-injected *chm^ru848^* showed a more preserved photoreceptor layer (white asterisk) with better retinal lamination and a thicker RPE layer compared to the uninjected mutants (*n* = 9 zebrafish per group). Scale bar: 50 μm; scale bar zoom in: 20 μm. (**c**) Cell viability is unchanged after *PFKM* overexpression in wt and *chm^ru848^* zebrafish (n = 9 zebrafish per group). Scale bar = 50 μm. ONL: outer nuclear layer; INL: inner nuclear layer. (**d**,**e**) Expression of anti-apoptotic (**d**) and pro-apoptotic (**e**) markers were analysed using RT-qPCR at 5 dpf in zebrafish eyes, from uninjected and injected wt and *chm^ru848^*. Data are expressed as mean ± SEM from n = 3 (with 20 eyes per group). Statistical significance was determined by one-way ANOVA. * *p* < 0.05; ** *p* < 0.005; **** *p* < 0.0001. “●”—wt; “▲”—wt + *PFKM*; “▼”—*chm^ru848^*; “◆”—*chm^ru848^* + *PFKM*.

**Figure 5 antioxidants-13-01587-f005:**
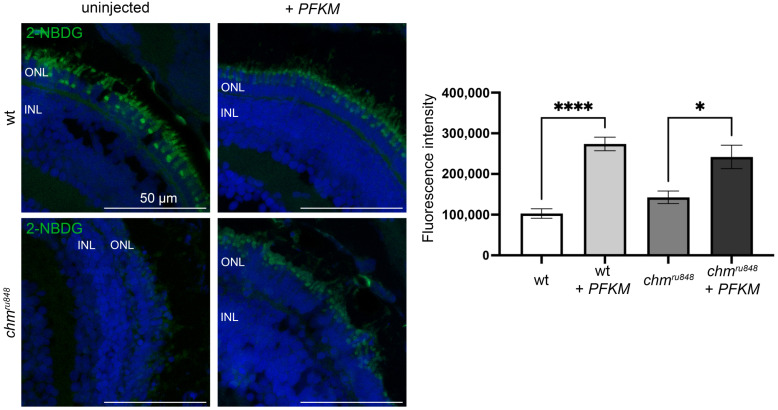
*PFKM* overexpression rescues 2-NBDG uptake in photoreceptors in *chm^ru848^* retina at 5 dpf. Scale bar = 50 μm. 2-NBDG is in green; nucleus stained with DAPI (blue). * *p* < 0.05; **** *p* < 0.0001.

**Figure 6 antioxidants-13-01587-f006:**
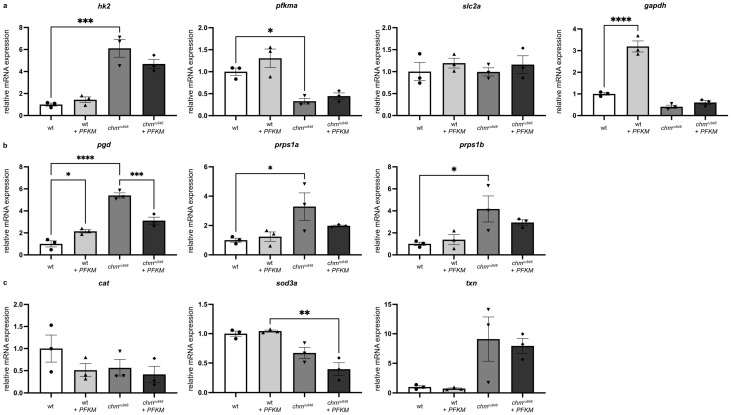
*PFKM* overexpression does not improve activity of the glycolysis pathway or oxidative stress 5 dpf in *chm^ru848^* zebrafish. Expression of genes involved in the glycolysis pathway (**a**), in the pentose phosphate oxidative branch (**b**) and in oxidative stress (**c**) were analysed using RT-qPCR at 5dpf in zebrafish eyes, from uninjected and injected wt and *chm^ru848^*. Data are expressed as mean ± SEM from n = 3 (with 20 eyes per group). Statistical significance was determined by One-way ANOVA. * *p* < 0.05; ** *p* < 0.005; *** *p* < 0.001; **** *p* < 0.0001. “●”—wt; “▲”—wt + *PFKM*; “▼”—*chm^ru848^*; “◆”—*chm^ru848^* + *PFKM*.

**Table 1 antioxidants-13-01587-t001:** Zebrafish and human primer sequences.

Gene	Forward Primer	Reverse Primer
*hPFKM* NM_000289.6	ATGACCCATGAAGAGCACCA	GCACCGGTGAAGATACCAAC
*cflara* NM_001313772.1	TGAAAGGACATGAGAGAAATGTGC	ATGGAGTGGTTTGTGTTGTGTTG
*mcl-1a* NM_131599.1	GCGATACTCGGCAGCTCTTA	GAAGACGCTGGATCATTCCTTT
*casp8* NM_131510.2	CAGAGACCAGGAACAAGGAGG	TAATTGTGCCAGCCGAAGAGT
*ddit3* NM_001082825.1	CAGCTGAACAATGGTTAACATGA	AATCAAGTTTGAATGTGAGTTGTTG
*hk2* NM_213066.1	AAACCACCCAGAGTTTGCTC	AGACGCAGTGTGTCCAGAAC
*pfkma* NM_001004575.2	AATACCATCACCACGACCTGT	GGTAACCGCAGTATCCTCCC
*slc2a1b* XM_002662528.6	TGATGGAAGGCGGAAAGCAAT	ACAGACAGAGACCACAGGG
*gapdh* NM_001115114.1	CGTCTTGAGAAACCTGCCAAG	AACCTGGTGCTCCGTGTATC
*pgd* NM_213453.2	GAGTTCGGCTGGTCTCTGAA	ATCTCGTGTCTGTACCCGTC
*prps1a* NM_001359894.1	CCGGTGGAGCAAAGAGAGTG	CACTCTGTCCTTCACGTCCC
*prps1b* NM_001076568.2	AGGAGCCAAGAGGGTTACCT	TCTCCAACCAGAACCATGCG
*cat* NM_130912.2	ACGATGACAACGTGACCCAA	CCATCAGGTTTTGCACCATGC
*sod3a* NM_001099236.1	TCAAGTGCGTGCCATCCATA	CCGCCGGATAAGTCCTTGTT
*txnb* NM_001002461.1	GACCATCGGGCCGTACTTTA	CATAAAGCGGCCACATCCTGT
*b-actin* NM_181601.5	CGAGCTGTCTTCCCATCCA	TCACCAACGTAGCTGTCTTTCTG

## Data Availability

The RNA-seq data generated by this study have been deposited in the NCBI Gene Expression Omnibus under the access code GSE254948, including unprocessed FASTQ files and associated gene count matrices.

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
