# Peer review of "Oxidative Stress, Inflammation and Altered Glucose Metabolism Contribute to the Retinal Phenotype in the Choroideremia Zebrafish"

_antioxidants, 2024, doi:10.3390/antiox13121587_

Round 1
Reviewer 1 Report
Summary:
The manuscript under review present research findings from a study conducted to demonstrate the relationship between retinal phenotype and oxidative stress, inflammation and altered glucose metabolism in a chmru848 zebrafish choroideremia model.
The study design is well-detailed, current and intriguing, clearly and effectively presenting the results, utilizing tables and figures to enhance data comprehension. The manuscript is written in high-quality scientific English
Although the study can serve as a valuable contribution in this field, some minor revisions are suggested to enhance the clarity, consistency, and flow.
Comments:
The Introduction section would benefit from a more detailed description of the characteristics of choroideremia, including specific features of this X-linked inherited retinal disease.
The formatting of the TXNIP abbreviation in lines 387 and 389 is inconsistent with its earlier usage.
Overall, the manuscript is well-written, and the suggested revisions are intended to improve its precision, depth, and relevance to the field.
Summary:
The manuscript under review present research findings from a study conducted to demonstrate the relationship between retinal phenotype and oxidative stress, inflammation and altered glucose metabolism in a chmru848 zebrafish choroideremia model.
The study design is well-detailed, current and intriguing, clearly and effectively presenting the results, utilizing tables and figures to enhance data comprehension. The manuscript is written in high-quality scientific English
Although the study can serve as a valuable contribution in this field, some minor revisions are suggested to enhance the clarity, consistency, and flow.
Comments:
The Introduction section would benefit from a more detailed description of the characteristics of choroideremia, including specific features of this X-linked inherited retinal disease.
The formatting of the TXNIP abbreviation in lines 387 and 389 is inconsistent with its earlier usage.
Overall, the manuscript is well-written, and the suggested revisions are intended to improve its precision, depth, and relevance to the field.
Reviewer 2 Report
This manuscript presents a compelling investigation into the mechanisms underlying retinal pathology in a zebrafish model of choroideremia (CHM). The study elucidates the interplay between oxidative stress, inflammation and altered glucose metabolism, providing valuable insights into the molecular basis of the disease. The work is well designed, using transcriptomic analysis, metabolic assays and gene overexpression experiments to identify potential therapeutic targets. However, some areas would benefit from further clarification and discussion.
I have made some comments about things I think you could do to improve your work. This does not mean that you have to agree or rewrite in the same way. It is just a suggestion and a different view with the aim of contributing.
1) While the introduction provides a clear overview of CHM, it lacks detail on the rationale for focusing on glucose metabolism. Including background on the role of glucose in retinal health and how its dysregulation may exacerbate CHM pathology would strengthen the narrative;
2) How might impaired glycolysis affect photoreceptor survival?
3) Could therapies targeting the oxidative pentose phosphate pathway exacerbate or alleviate oxidative stress?
4) Please, address potential reasons why PFKM overexpression, despite partial rescue, did not alter glycolysis and oxidative stress markers.
5) Please, include a brief comparison with other metabolic therapies under investigation for retinal dystrophies.
The manuscript makes a significant contribution to the understanding of the pathogenesis of CHM, but addressing the above points will enhance its clarity and impact. The innovative approach and promising findings of the study make it suitable for publication with minimal revisions.
-
